# Electrocardiogram, Echocardiogram and NT-proBNP in Screening for Thromboembolism Pulmonary Hypertension in Patients after Pulmonary Embolism

**DOI:** 10.3390/jcm11247369

**Published:** 2022-12-12

**Authors:** Olga Dzikowska-Diduch, Katarzyna Kurnicka, Barbara Lichodziejewska, Iwona Dudzik-Niewiadomska, Michał Machowski, Marek Roik, Małgorzata Wiśniewska, Jan Siwiec, Izabela Magdalena Staniszewska, Piotr Pruszczyk

**Affiliations:** 1Department of Internal Medicine & Cardiology, Medical University of Warsaw, Lindleya 4, 02-005 Warsaw, Poland; 21st Department of Radiology, Medical University of Warsaw, Lindleya 4, 02-005 Warsaw, Poland

**Keywords:** screening after pulmonary embolism, chronic thromboembolic pulmonary disease, chronic thromboembolic pulmonary hypertension, diagnostic work-up of post-pulmonary syndrome

## Abstract

Background: The annual mortality of patients with untreated chronic thromboembolism pulmonary hypertension (CTEPH) is approximately 50% unless a timely diagnosis is followed by adequate treatment. In pulmonary embolism (PE) survivors with functional limitation, the diagnostic work-up starts with echocardiography. It is followed by lung scintigraphy and right heart catheterization. However, noninvasive tests providing diagnostic clues to CTEPH, or ascertaining this diagnosis as very unlikely, would be extremely useful since the majority of post PE functional limitations are caused by deconditioning. Methods: Patients after acute PE underwent a structured clinical evaluation with electrocardiogram, routine laboratory tests including NT-proBNP and echocardiography. The aim of this study was to verify whether the parameters from echocardiographic or perhaps electrocardiographic examination and NT-proBNP concentration best determine the risk of CTEPH. Results: Out of the total number of patients (*n* = 261, male *n* = 123) after PE who were included in the study, in the group of 155 patients (59.4%) with reported functional impairment, 13 patients (8.4%) had CTEPH and 7 PE survivors had chronic thromboembolic pulmonary disease (CTEPD) (4.5%). Echo parameters differed significantly between CTEPH/CTEPD cases and other symptomatic PE survivors. Patients with CTEPH/CTEPD also had higher levels of NT-proBNP (*p* = 0.022) but concentration of NT-proBNP above 125 pg/mL did not differentiate patients with CTEPH/CTEPD (*p* > 0.05). Additionally, the proportion of patients with right bundle brunch block registered in ECG was higher in the CTEPH/CTED group (23.5% vs. 5.8%, *p* = 0.034) but there were no differences between the other ECG characteristics of right ventricle overload. Conclusions: Screening for CTEPH/CTEPD should be performed in patients with reduced exercise tolerance compared to the pre PE period. It is not effective in asymptomatic PE survivors. Patients with CTEPH/CTED predominantly had abnormalities indicating chronic thromboembolism in the echocardiographic assessment. NT-proBNP and electrocardiographic characteristics of right ventricle overload proved to be insufficient in predicting CTEPH/CTEPD development.

## 1. Introduction

Chronic thromboembolism pulmonary hypertension (CTEPH) is a rare disease but, unlike other forms of pulmonary hypertension, is curable [1]. Untreated CTEPH significantly affects the patient’s prognosis [2]. The annual mortality in patients with untreated pulmonary hypertension is approximately 50% unless a timely diagnosis is followed by adequate treatment [3]. Early diagnosis of CTEPH is nonetheless essential but continues to be a huge challenge because there are no specific signs or symptoms of CTEPH. Moreover, symptoms of pulmonary hypertension develop slowly and are usually explained by more common causes. It has been estimated that the majority of CTEPH diagnoses nowadays still have a diagnostic delay by well over 1 year [4]. Chronic thromboembolic pulmonary disease (CTEPD) describes patients with chronic thromboembolic occlusions of pulmonary arteries but without PH at rest; however, change in the definition of PH with a decrease in the threshold mean pulmonary artery pressure from 25 to 21 mmHg may influence the designation of former CTEPD as CTEPH patients. Moreover, PE survivors with CTEPD benefit from the same treatment, including pulmonary endarterectomy, balloon pulmonary angioplasty and lifelong anticoagulation. Most cases of CTEPD/CTEPH occur in patients with a history of pulmonary embolism (PE) or/and deep vein thrombosis; therefore, screening for pulmonary hypertension seems reasonable in these patients.

In PE survivors with functional limitation, the diagnostic work-up starts with echocardiography, followed by ventilation/perfusion lung scintigraphy and right heart catheterization (RHC) with pulmonary angiography [5]. RHC is the gold standard for the diagnosis of pulmonary hypertension [2]; however, noninvasive tests providing diagnostic clues to confirm CTEPH or ascertain this diagnosis as very unlikely would be extremely useful, since the majority of post PE functional limitations are caused by deconditioning.

The European Society of Cardiology (ESC) guidelines propose echocardiography at rest as an initial examination, when CTEPH is suspected based on patient’s clinical presentation [2,5]. Echo evaluations include estimating peak velocity of tricuspid valve regurgitation, calculation of atrioventricular pressure gradients and detection of indirect signs of pulmonary hypertension which should aim to estimate a level of probability of pulmonary hypertension. However, the accuracy of echocardiography that provides clues to the presence or absence of CTEPH is quite high (sensitivity of 70–100% and specificity of 72–89%) but echocardiography, performed according to the European Association of Cardiovascular Imaging (EACVI) guidelines, is not widely available and is associated with overdiagnosis and cost-ineffectiveness [6]. N-terminal pro-brain natriuretic peptide (NT-proBNP) levels correlate with myocardial dysfunction and can be elevated in case of almost any heart disease. NT-proBNP has been evaluated to stratify risk in patients with acute PE and remains the only biomarker that seems to be a strong predictor of prognosis and therefore is widely used in the routine practice in PH centers [2]. Its sensitivity and specificity in the diagnosis of CTEPH are around 82% and 70%, respectively [7]. Electrocardiogram (ECG) abnormalities including P pulmonale, right axis deviation (RAD), right ventricle (RV) hypertrophy and right bundle branch block (RBBB) are more common in severe PH than in the mild elevation of pulmonary pressure [2]. Normal ECG does not exclude the diagnosis of PH. Klok et al. showed in two independent cohort studies that the combination of a normal NT-proBNP level and the absence of specific ECG characteristics of RV overload accurately differentiates patients after PE with PH from those without PH with a sensitivity of 94–100% [7,8]. Moreover, the InShape II study showed that an algorithm including clinical probability of CTEPH, ECG and NT-proBNP levels can accurately exclude CTEPH, without the need for echocardiography (Boon InShape II) [8].

The aim of our study was to verify whether the parameters from echocardiographic or perhaps electrocardiographic examination and NT-proBNP concentration best determine the risk of CTEPH.

## 2. Methods

This is a post hoc analysis of a prospectively followed cohort of patients after pulmonary embolism. Patients were eligible for inclusion if aged 18 years or older and had a computed tomography pulmonary angiography proving diagnosis of symptomatic acute PE, and had been treated with therapeutically dosed anticoagulant therapy for at least 3 months according to current guidelines. After the discharge, all patients underwent, as previously reported, standard outpatient follow-up for at least 6 months following the acute PE event and were anticoagulated for at least 6 months. Briefly, we included all consecutive patients after acute PE with the exception of subjects with comorbidities significantly limiting survival or mobility (patients with advanced cancer or bed ridden subjects). At follow-up, all subjects underwent a structured clinical examination focused on their functional limitation. All patients were evaluated for the presence of exertional dyspnea, effort angina, exercise-limiting palpitations and a reduced exercise tolerance. In all patients ECG and routine laboratory tests were analyzed including hemoglobin, estimated glomerular filtration rate and NT-proBNP. All patients underwent a structured evaluation with echocardiography, which were performed by an experienced cardiologist according to the current EACVI recommendations [9]. Subjects with at least intermediate echocardiographic probability of PH according to the ESC guidelines [2] were referred to the detailed complete work-up for CTEPH.

### 2.1. Echocardiography

All echocardiograms were performed with Philips IE33 or Epic 7 according to the predefined standardized protocol by an experienced cardiologist, and focused on echocardiographic criteria for suspected PH (increased systolic peak tricuspid regurgitation velocity, dilated right ventricle, flattened interventricular septum, distended inferior vena cava with diminished inspiratory collapsibility or enlarged right atrial area) according to the 2015 ESC guidelines [2]. However, detailed evaluation of left ventricular (LV) morphology and function was also performed. The examinations were digitally recorded, and reviewed, when necessary. Patients were examined in the left lateral position. The dimensions of the right and left ventricles were measured in the parasternal long-axis view and apical four chamber view (4C) at the level of the mitral and tricuspid valve tips in late diastole defined by the R wave of continuous ECG tracing [10]. Tricuspid valve regurgitation was qualitatively assessed with color Doppler and peak gradient (tricuspid regurgitation peak gradient—TRPG) was calculated by simplified Bernoulli’s formula after using tricuspid regurgitant flow peak velocity. The examination was completed by the measurement of the inferior vena cava (IVC) at late expiration. In the parasternal short axis view flattening of the interventricular septum was assessed qualitatively, and acceleration time (AcT) of pulmonary ejection was measured in the RV outflow tract, just below the pulmonary valve. Measurements were averaged over 3 consecutive heart cycles. In M-mode presentation, RV function was assessed by tricuspid annular plane systolic excursion (TAPSE) measurement. We measured the distance (mm) of systolic excursion of the RV annular segment along its longitudinal plane, from a standard apical 4-chamber view. The left ventricular ejection fraction was calculated according to Simpson’s formula employing a two-dimensional image of the LV chamber during systole and diastole in the four and two chamber apical views.

### 2.2. Electrocardiogram

Conventional 12-lead ECGs will be recorded with the patient in supine position for a 10-s period using the standard 12-lead electrode configuration at a conventional speed (25 mm/s) and sensitivity (1 mV/10 mm). Sinus rhythm or arrhythmias and heart rate were registered. ECGs were also evaluated for the presence or absence of the following criteria that had been reported to occur more commonly in PH: right axis deviation (RAD defined as dominant S wave lead I with dominant R wave leads II and III), right bundle branch block (RBBB defined as QRS duration >120 ms with rSR pattern V1–V3), S1Q3T3 pattern (the presence of S waves in lead I and Q waves in lead III, each with amplitudes> 1.5 mm in association with negative T waves in lead III).

### 2.3. NT-proBNP

NT-proBNP levels were determined with the use of a quantitative immunoassay. Age- and gender-dependent thresholds for normal values as determined by the respective manufacturers were used.

### 2.4. Diagnoses

CTEPH was diagnosed when the invasive right heart catheterization showed that the mean pulmonary artery pressure was ≥25 mmHg at rest, pulmonary wedge pressure was ≤ 15 mmHg and abnormal imaging findings on the ventilation/perfusion lung scan, while CTEPD was diagnosed as CTEPH when mean pulmonary artery pressure, was< 25 mmHg [2].

### 2.5. Statisitcal Analysis

Statistical analysis was carried out in R software, version 4.0.5. Normality of distribution was verified using Shapiro–Wilk test and based on skewness and kurtosis values as well as visual assessment of histograms. Groups’ comparison was conducted with chi-squared test or Fisher’s exact test (nominal data) and with Welch *t*-test or Mann–Whitney U test (quantitative data), as appropriate. Additionally, we calculated odds ratio (OR) or mean/median differences (MD) between groups, including 95% confidence intervals.

In order to identify optimal cut-off points for each parameter as discriminator of CTEPH/CTED vs. healthy patients with symptoms, receiver operating characteristic (ROC) curves were created. Cut-off point calculation was based on Youden index, including measures of sensitivity, specificity, accuracy, negative predictive value (NPV) and positive predictive value (PPV).

## 3. Results

Out of the total number of patients (*n* = 261, male *n* = 123) after PE who were included in the study, 155 patients (59.4%) were with symptoms (mainly functional impairment) and 106 patients (40.6%) were without any symptoms. No significant differences were confirmed between both groups in sex, while age was significantly different—patients with symptoms were older than patients without symptoms, MD = 11.32 CI95 [7.41,6.21], *p* < 0.001.

In the group of patients with symptoms, 13 patients (8.4%) had CTEPH and 7 were survivors of CTEPD (4.5%) vs. no cases of CTEPH/CTEPD in the group without symptoms, *p* < 0.001. Chronic heart failure was recognized in about 50% of patients as the major cause of reduced exercise tolerance. Most of them presented a preserved ejection fraction. Other causes of functional limitation in the studied group included valve heart disease (6%), coronary artery diseases (6%), chronic obstructive pulmonary diseases (6%) and newly diagnosed permanent or paroxysmal atrial fibrillation in 6.4% of patients. Noncardiopulmonary pathologies including severe obesity in patients, newly diagnosed neoplasms or anemia contributed to decreased functional capacity in approximately 10% of symptomatic patients.

Echocardiogram, ECG and NT-proBNP were assessed 6 ± 0.97 months after PE.

Patients with symptoms had a significantly higher level of diameter of inferior vena cava (IVC) (*p* = 0.008), RV in 4 chamber view (RV4ch) (*p* = 0.002), elevated RV to LV ratio in 4 chamber view (*p* = 0.006), right atrium area (RAA) (*p* < 0.001), TRPG (*p* = 0.001), and left atrium area (LAA) (*p* < 0.001), than patients without functional impairment after PE. Indeed, there was a significant difference in LV EF% between the symptomatic and asymptomatic groups: 60.77 ± 5.43% vs. 62.91 ± 3.22%; *p* < 0.001. The LAA differed significantly between groups and was significantly elevated in symptomatic patients, which indicates that persistent dyspnea on exertion was also caused by the disease of the left heart, mainly left ventricular diastolic dysfunction. The AcT of pulmonary output average level was significantly lower (*p* = 0.001) in patients with functional limitation.

NT-proBNP concentrations were significantly higher in symptomatic patients than in the group without symptoms (*p* < 0.001). Additionally, the proportion of patients with NT-proBNP above norm was higher in symptomatic PE survivors (43.9% vs. 18.9% in group without symptoms), OR = 3.35 CI95 [1.82,6.34], *p* < 0.001. Interestingly, every fifth patient after PE without any deterioration of exercise tolerance had an increased concentration of NT-proBNP.

PE survivors with exercise intolerance had a higher level of d-dimer compared to patients who fully recovered functionally (*p* = 0.049) despite ongoing anticoagulation and no significant difference in drug taken.

Heart rhythm was also significantly different between both groups (*p* = 0.005). Patients with symptoms more often had atrial fibrillation (6.5% of cases vs. no cases in patients without symptoms) and less frequently had sinus rhythm (92.7% vs. 99.1%). No significant differences between both groups for drugs taken, echocardiographic LV4ch, TAPSE and ECG parameters (HR, RAD, RBBB, S1Q3T3) (Table 1.) were found.

As a second step, in the 155 patients with symptoms, patients with and without CTEPH/CTEPD were compared. Patients with CTEPH/CTEPD had a significantly higher level of: RV4ch (*p* = 0.003), RAA (*p* = 0.004), TRPG (*p* < 0.001) and NTproBNP (*p* = 0.022) than patients without CTEPH/CTEPD. For AcT of pulmonary ejection average level was significantly lower (*p* = 0.008) in patients with CTEPH/CTEPD. Additionally, the proportion of patients with RBBB was higher in the group with CTEPH/CTEPD (23.5% vs. 5.8% in the group without CTEPH/CTED), OR = 4.92 CI95 [1.90,24.18], *p* = 0.034. No differences between patients with and without CTEPH/CTEPD were confirmed for LAA, the proportion of RAD and S1Q3T3 in ECG and NTproBNP> 125 pg/mL (*p* > 0.05 in all cases) (Table 2 and Table 3).

Receiver operating characteristic curves as discrimination of CTEPH/CTEPD vs. symptomatic but without CTEPH/CTED were significant for RV4ch (*p* = 0.002), RAA (*p* = 0.001), TRPG (*p* < 0.001) and AcT (*p* = 0.001) with satisfactory or very good level of area under the curve (AUC); from AUC = 0.723 CI95 [0.591,0.855] for RV4ch to AUC = 0.868 CI95 [0.785,0.952] for TRPG. The level of optimal cut-off points for particular parameters with corresponding sensitivity and specificity is summarized in Table 4.

## 4. Discussion

Our analysis showed that screening for CTEPH should be given to symptomatic patients after PE. Although ECG, echo and NT-proBNP screening was performed in all of 261 patients included in our study, CTEPH and CTEPD was diagnosed only in group of patients with dyspnea on exertion. PE survivors who completely recovered functionally had no significant abnormalities in the echo, ECG and laboratory test. This is consistent with the observations of Held et al. who suggested focusing diagnostic procedures on only symptomatic patients [11]. Habib and Torbicki also said in 2010 that echocardiographic screening for CTEPH is not effective in asymptomatic patients [12].

The echocardiographic estimation of the likelihood of PH is among the key elements in the decision-making process by identifying patients for whom RHC is warranted, facilitating earlier diagnosis and earlier medical management [13]. A meta-analysis calculated the accuracy of echocardiography vs. RHC for PH diagnosis and found a sensitivity of 83% (95% CI, 73–90%) and specificity of 72% (95% CI, 53–85%) and that echocardiography has been shown to miss PH in as many as 10–31% of cases [14]. Indeed, in our analysis, significant echocardiographic abnormalities were assessed in CTEPH/CTEPD group, but the diagnosis of chronic thromboembolic pulmonary artery occlusion explaining exercise intolerance was made in only 20 of 155 patients who complained of functional limitation. It cannot be ruled out that some cases of CTEPH/CTEPD had been missed. Patients with CTEPH/CTEPD had typical echocardiographic signs of pulmonary hypertension included enlargement of the right atrium and right ventricle, elevated RV to LV ratio in the four chamber view and significant elevated TRPG, IVC diameter and RVSP. Our findings are consistent with previous observations of Habib, Torbicki and Surinder Janda et al. [12,14]. In our study, the AcT of pulmonary output was significantly lower in patients with CTEPH/CTEPD compared to other symptomatic PE survivors, as in the analysis of Kitabatake et al. [15]. Moreover, our echocardiographic assessment after PE revealed a significant elevated left atrium area in patients suffering from dyspnea on exertion, which suggests that the functional limitation is also due to left heart disease, mainly diastolic dysfunction [16]. Chronic heart failure with preserved systolic function, which is much more common than chronic pulmonary artery occlusion, may explain elevated concentrations of NT-proBNP in symptomatic patients after PE [17,18]. Survivors with symptoms were also much older and more likely to have fibrillation than asymptomatic patients, which also explains the increased concentration of BNP [19]. Obviously, mean concentrations of BNP were higher in CTEPH/CTEPD but levels above 125 pg/mL were exceeded similarly frequently in symptomatic patients with and without chronic thromboembolism. NT-proBNP allows for only sufficient differentiation of patients with CTEPH/CTEPD (AUC 0,659, Figure 1). In general, conventional ECG criteria had low diagnostic accuracy for the presence of increased RV afterload [20,21]. In our study, the proportion of patients with RBBB was higher in groups with CTEPH/CTEPD (23.5% vs. 5.8%), *p* = 0.034, but there were no significant differences in ECG characteristics of right ventricle overload as RAD and S1Q3T3. The combination of ECG and NT-proBNP in the Leiden CTEPH rule-out criteria may be useful in diagnostic after PE but in in the group of 261 patients we studied, they did not allow to safely and effectively exclude PH or indicate patients for further invasive work-up [22]. Meaningful screening programs should be simple, widely available and non-invasive. However, diagnostic tests should quickly and clearly indicate which patient will benefit from further work-up [23,24]. Since symptoms initially in CTEPH appear during exercise, the tests performed at rest, including electrocardiogram, echocardiogram or RHC, may lack sensitivity. Although prospective evaluation of larger cohorts is still lacking, functional tests are a promising complementary diagnostic tool for functional evaluation of patients with chronic pulmonary vascular disease [7,25].

## 5. Conclusions

Screening for CTEPH/CTEPD should be performed in patients with reduced exercise tolerance compared to the pre PE period, and it is not effective in asymptomatic PE survivors. Patients with CTEPH/CTED had presented predominantly abnormalities indicating chronic thromboembolism in the echocardiographic assessment. NT-proBNP and electrocardiographic characteristics of right ventricle overload proved to be insufficient in predicting CTEPH/CTEPD development.

## Figures and Tables

**Figure 1 jcm-11-07369-f001:**
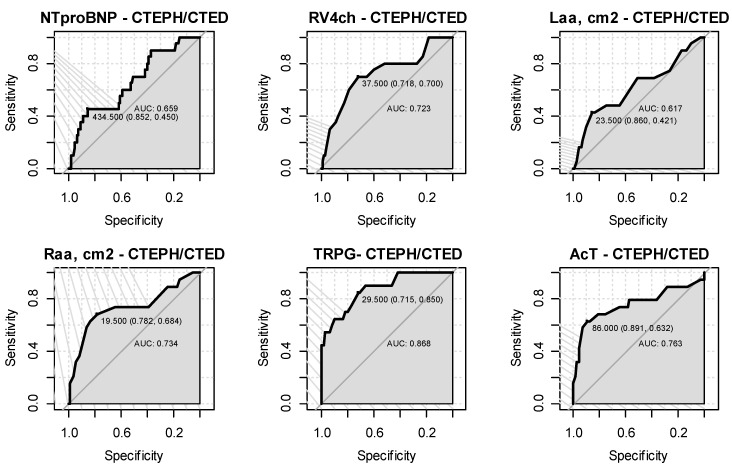
ROC curves for particular parameters as diagnostic test for CTEPH/CTEPD vs. patients with symptoms but without CTEPH/CTEPD (numbers of each chart include AUC value as well as optimal cut-off point with specificity and sensitivity values).

**Table 1 jcm-11-07369-t001:** Comparison of patients after pulmonary embolism with and without symptoms.

Characteristics	*n*	Patients with Symptoms	*n*	Patients without Symptoms	MD/OR (95% CI)	*p*
*n*	155		106			
Sex, female, *n* (%)	155	85 (54.8)	106	53 (50.0)	1.21 (0.72,2.05)	0.52
Age, years, mean ± SD	155	61.07 ± 17.10	106	49.75 ± 18.36	11.32 (7.4,16.21)	<0.001 ^2^
CTEPH/CTED, *n* (%)	155	20 (12.9)	106	0 (0.0)	-	<0.001 ^1^
Drug, *n* (%)						
Acenocumarol	152	52 (34.2)	86	37 (43.0)	-	0.0651
Dabigatran		10 (6.6)		4 (4.7)		
Dalteparin		1 (0.7)		1 (1.2)		
Enoxaparin		8 (5.3)		5 (5.8)		
Nadroparin		2 (1.3)		0 (0.0)		
Riwaroxaban		64 (42.1)		22 (25.6)		
Warfarin		15 (9.9)		17 (19.8)		
IVC, mean ± SD	151	15.36 ± 4.60	104	13.96 ± 3.76	1.40 (0.37,2.44)	0.0082
LV4ch, mean ± SD	131	45.67 ± 5.50	73	43.50 ± 4.24	2.17 (−1.23,1.87)	0.6832
RV4ch, mean ± SD	130	33.50 ± 6.60	74	31.88 ± 2.59	1.63 (0.85,3.86)	0.0022
RV/LV, mean ± SD	130	0.80 ± 0.12	73	0.75 ± 0.12	0.04 (0.01,0.08)	0.0062
LV EF, mean ± SD	155	60.77 ± 5.43	106	62.91 ± 3.22	−2.14 (−3.19,01.07)	<0.001 ^2^
LAA, cm^2^, mean ± SD	140	19.89 ± 4.07	81	16.96 ± 3.31	2.92 (1.89,3.88)	<0.001 ^2^
RAA, cm^2^, mean ± SD	138	18.12 ± 4.14	78	15.42 ± 3.28	2.71 (1.67,3.68)	<0.001 ^2^
Heart rhythm, *n* (%)						
Atrial fibrillation	154	10 (6.5)	106	0 (0.0)	-	0.0051
Stimulation		1 (0.6)		0 (0.0)		
Tachycardia		0 (0.0)		1 (0.9)		
Sinus rhythm		143 (92.7)		105 (99.1)		
TRPG, median (Q1,Q3)	150	25.50 (20.00,32.75)	103	23.00 (17.00,27.50)	2.50 (1.00,6.00)	0.0013
RVSP, median (Q1,Q3)	146	31.00 (25.00,40.75)	102	28.00 (23.00,33.75)	3.00 (2.00,7.00)	0.0013
AcT, mean ± SD	148	112.08 ± 28.07	101	123.70 ± 24.03	−11.62 (−18.17,−5.07)	0.0012
TAPSE, mean ± SD	134	23.34 ± 3.79	80	23.66 ± 3.21	−0.33 (−1.29,0.63)	0.5032
HR, mean ± SD	137	70.58 ± 10.66	58	69.45 ± 9.71	1.14 (−1.97,4.24)	0.4702
RAD, *n* (%)	131	4 (3.1)	57	0 (0.0)	-	0.3161
RBBB, *n* (%)	121	10 (8.3)	53	1 (1.9)	4.65 (0.63,206.97)	0.1761
S1Q3T3, *n* (%)	117	26 (22.2)	53	7 (13.2)	1.87 (0.72,5.50)	0.243
NTproBNP, median (Q1,Q3) (pg/mL)	155	108.00 (45.00,339.50)	106	29.00 (20.00,96.25)	79.00 (31.00,85.00)	<0.001 ^3^
NTproBNP > 125 pg/mL, *n* (%)	155	68 (43.9)	106	20 (18.9)	3.35 (1.82,6.34)	<0.001
D-dimer,(ng/mL) median (Q1,Q3)	95	300.00 (205.50,488.50)	95	239.00 (170.00,420.00)	61.00 (0.01,93.00)	0.0493

AcT—acceleration time of pulmonary ejection, CTEPD—chronic thromboembolic pulmonary disease, CTEPH—chronic thromboembolic pulmonary hypertension, HR—heart rate, LV—left ventricle, 4ch—four chamber view, NT-proBNP: N-terminal pro-brain natriuretic peptide, RAD—right axis deviation, RBBB—right heart catheterization, RV—right ventricle, TRPG—tricuspid regurgitation peak gradient, TAPSE—tricuspid annular plane systolic excursion, RVSP—right ventricle systolic pressure, MD—mean/median difference between groups calculated as patients with symptoms minus patients without symptoms with 95% confidence interval, OR—odds ratio between both groups, with 95% confidence interval. Groups compared with chi-square test or Fisher exact test ^1^ for nominal data with *t*-test ^2^ or Mann–Whitney U test ^3^ for continuous data.

**Table 2 jcm-11-07369-t002:** Comparison of patients after pulmonary embolism with and without CTEPH/CTED among symptomatic patients.

Characteristics.	*n*	Patients with Symptoms with CTEPH/CTEPD	*n*	Patients with Symptoms without CTEPH/CTEPD	MD/OR (95% CI)	*p*
*n*	20		135			
RV4ch, mean ± SD	20	36.50 ± 7.78	110	32.00 ± 6.58	4.50 (1.61,6.99)	0.003 ^2^
LAA, cm^2^, mean ± SD	19	21.58 ± 4.68	121	19.62 ± 3.93	1.96 (−0.39,4.31)	0.097 ^2^
RAA, cm^2^, mean ± SD	19	21.53 ± 5.08	119	17.58 ± 3.71	3.95 (1.42,6.47)	0.004 ^2^
TRPG, median (Q1,Q3)	20	45.00 (30.75,62.00)	130	24.00 (20.00,30.00)	21.00 (13.00,33.00)	<0.001 ^3^
AcT, mean ± SD	19	88.42 ± 39.21	129	115.57 ± 24.36	−27.15 (−46.43,−7.86)	0.008 ^2^
RAD, *n* (%)	19	1 (5.3)	112	3 (2.7)	2.01 (0.04,26.60)	0.469 ^1^
RBBB, *n* (%)	17	4 (23.5)	104	6 (5.8)	4.92 (1.90,24.18)	0.034 ^1^
S1Q3T3, *n* (%)	16	6 (37.5)	101	20 (19.8)	2.41 (0.64,8.40)	0.191
NTproBNP, median (Q1,Q3)	20	151.00 (85.25,843.00)	135	99.00 (43.00,300.50)	52.00 (12.00,372.00)	0.022 ^3^
NTproBNP > 125, *n* (%)	20	12 (60.0)	135	56 (41.5)	2.11 (0.74,6.36)	0.149

AcT: acceleration time of pulmonary ejection, CTEPD: chronic thromboembolic pulmonary disease, CTEPH: chronic thromboembolic pulmonary hypertension, LAA—left atrium area, NT-proBNP: N-terminal pro-brain natriuretic peptide, RAA—right atrium area, RAD—right axis deviation, RBBB—right heart catheterization, RV4ch—right ventricle 4 chamber view, TRPG—tricuspid regurgitation peak gradient, MD—mean/median difference between groups calculated as patients with CTEPH/CTED minus patients without CTEPH/CTED with 95% confidence interval, OR—odds ratio between both groups, with 95% confidence interval. Groups compared with chi-square test or Fisher exact test ^1^ for nominal data, with *t*-test ^2^ or Mann–Whitney U test ^3^ for continuous data.

**Table 3 jcm-11-07369-t003:** Comparison of patients after pulmonary embolism with CTEPH and CTED among symptomatic patients.

Characteristics	*n*	Patients with CTEPH	*n*	Patients with CTED	MD / OR (95% CI)	*p*
*n*	13		7			
RV4ch, mean ± SD	13	42.00 (42.00,42.00)	7	31.00 (31.00,31.00)	11.00 (−4.00,18.00)	0.249 ^3^
LAA, cm^2^, mean ± SD	13	21.92 ± 4.70	6	20.83 ± 5.00	1.09 (−4.36,6.54)	0.663 ^2^
RAA, cm^2^, mean ± SD	13	22.77 ± 4.90	6	18.83 ± 4.75	3.94 (−1.33,9.21)	0.127 ^2^
TRPG, median (Q1,Q3)	13	59.23 ±23.81	7	34.71 ± 12.66	24.52 (7.39,41.65)	0.008 ^2^
AcT, mean ± SD	12	78.50 (67.50,86.25)	7	106.00 (78.50,140.00)	−27.50 (−72.00,10.00)	0.162 ^3^
RAD, *n* (%)	12	1 (8.3)	7	0 (0.0)	-	>0.999 ^1^
RBBB, *n* (%)	10	3 (30.0)	7	1 (14.3)	2.44 (0.15,156.95)	0.603 ^1^
S1Q3T3, *n* (%)	10	3 (30.0)	6	3 (50.0)	0.45 (0.03,5.51)	0.607 ^1^
NTproBNP, median (Q1,Q3)	13	435.00 (132.00,1494.00)	7	107.00 (76.50,313.50)	328.00 (−33.00,1387.00)	0.115 ^3^
NTproBNP > 125, *n* (%)	13	10 (76.9)	7	2 (28.6)	7.32 (0.74,117.26)	0.062 ^1^

AcT: acceleration time of pulmonary ejection, CTEPD: chronic thromboembolic pulmonary disease, CTEPH: chronic thromboembolic pulmonary hypertension, LAA—left atrium area, NT-proBNP: N-terminal pro-brain natriuretic peptide, RAA—right atrium area, RAD-right axis deviation, RBBB-right heart catheterization, RV4ch—right ventricle 4 chamber view, TRPG-tricuspid regurgitation peak gradient, MD—mean/median difference between groups calculated as patients with CTEPH minus patients without CTEPD with 95% confidence interval, OR—odds ratio between both groups, with 95% confidence interval. Groups compared with Fisher exact test ^1^ for nominal data, with *t*-test ^2^ or Mann-Whitney U test ^3^ for continuous data.

**Table 4 jcm-11-07369-t004:** Results for measurement of different parameters in the diagnosis of CTEPH/CTEPD in symptomatic pulmonary embolism survivors.

Characteristics	AUC (95% CI)	Cut-off Point	Sensitivity	Specificity	Accuracy	NPV	PPV	*p*
RV4ch	0.723 (0.591,0.855)	37.5	0.7	0.72	0.72	0.93	0.31	0.002
LAA, cm^2^	0.617 (0.464,0.771)	23.5	0.42	0.86	0.8	0.9	0.32	0.056
RAA, cm^2^	0.734 (0.589,0.879)	19.5	0.68	0.78	0.77	0.94	0.33	0.001
TRPG	0.868 (0.785,0.952)	29.5	0.85	0.72	0.73	0.97	0.31	<0.001
AcT	0.763 (0.611,0.914)	86	0.63	0.89	0.86	0.94	0.46	0.001
NTproBNP	0.659 (0.528,0.789)	434.5	0.45	0.85	0.8	0.91	0.31	0.297

AcT: acceleration time of pulmonary ejection, AUC—area under the curve with 95% confidence interval (CI), CTEPD—chronic thromboembolic pulmonary disease, CTEPH—chronic thromboembolic pulmonary hypertension, LAA—left atrium area, NPV—negative predictive value, NT-proBNP—N-terminal pro-brain natriuretic peptide, PPV—positive predictive value, RAA—right atrium area, TRPG—tricuspid regurgitation peak gradient.

## Data Availability

The study did not report any data.

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
