# Peer review of "Electrocardiogram, Echocardiogram and NT-proBNP in Screening for Thromboembolism Pulmonary Hypertension in Patients after Pulmonary Embolism"

_jcm, 2022, doi:10.3390/jcm11247369_

Round 1

Reviewer 1 Report

Thank you for allowing me to review your article. the ability to predict which patients with PE will  go on to develop CTPH/CTPD has a significant impact on patient care. The authors accrued valuable echocardiographic, ACD and BNP data that clearly answers their hypothesis 

If i may ask that you clarify some questions:

1.     How was functional capacity or exercise tolerance assessed during the follow-up? Was it a subjective method or was there objective data (example 6-minute walk test) ?

2.     What kind of PE did the patients have? Were these provoked (catheter related / surgery) or unprovoked? If provoked why were thye on AC for more than 3 months? Was it because of the study or is that common practice?

3.     Line 63 : Can you  elaborate more on this? How does a PE lead to deconditioning? Do you mean that because of the decreased exercise tolerance patients become deconditioned?

4.     In line 103 : Please elaborate on what this work up required by the ESC guidelines

5.     Line 106 : Can you elaborate what echo findings you were looking at (I know these are mentioned later in the manuscript but for the reader to know preemptively what you are looking for is helpful)

6.     Line 77-78 “Its sensitivity in diagnosis of CTEPH is around 82% and specificity of 70%” Can you place a reference here

7.     Line 79-81: “Electrocardiogram (ECG) abnormalities included P pulmonale, right axis deviation (RAD), right ventricle (RV) hypertrophy, right bundle branch block (RBBB) are more common in severe PH than in mild elevation of pulmonary pressure” Can you place a reference here?

8.     Line 167 : Can you define what an elevated RV/LV ratio is ?

9.     Line 168 TRPG, was not a defined abbreviation anywhere in the manuscript (except in the tables)

10.  Was there any difference in LV EF between the symptomatic and asymptomatic groups, I think that would be an important value to add in table 1

11. Table 1: the word “stimulation” is misspelled.

12. Table 1+2 , I am not sure that you need the extra 2 columns for “n”

13.  In your discussion can you comment on the theorized effect of afib and age on the difference in Pro BNP between the two groups?

14.  In your discussion can you comment on how you came to the conclusion that their decrease in exercise capacity was sec to CTPH/PD versus other L sided issues especially since these patients had much higher rates of afib and possibly diastolic dysfunction

15. How many symptomatic patients get a R heart cath and what were the results of the ones that did?

Author Response

Reviewer 1

Comment 1:     How was functional capacity or exercise tolerance assessed during the follow-up? Was it a subjective method or was there objective data (example 6-minute walk test) ?

Reply 1: The initial stage of pulmonary embolism follow up was performed by   managing physician. Checkup included assessment of the patient's physical exercise tolerance compared to pre PE period.  Further diagnosis included echocardiography and 6 minute walk test was carried out in patients reporting a reduction in physical activity.

Comment 2:     What kind of PE did the patients have? Were these provoked (catheter related / surgery) or unprovoked? If provoked why were they on AC for more than 3 months? Was it because of the study or is that common practice?

Reply 2:  More than half (56%) of the patients were diagnosed with pulmonary embolism without an identifiable risk factor. However  the access to the ambulatory clinic is  rather limited and  some patient   have the first control  visit slightly later than 3  months after acute pulmonary embolism

Comment 3.     Line 63 : Can you elaborate more on this? How does a PE lead to deconditioning? Do you mean that because of the decreased exercise tolerance patients become deconditioned?

Reply 3: Patients with acute pulmonary embolism usually have limited exercise tolerance, and even in some cases they are reluctant to undertake activity and therefore do not engage in physical activity.  Moreover in some cases that they avoid exercise for fear of a symptoms reccurence. In many countries, including Poland, rehabilitation programs for patients after pulmonary embolism have not been developed yet and implemented so far. Lack of physical activity causes  unfavorable changes in muscles and further deterioration. It seems that muscle deconditioning is largely responsible for the frequently reported exercise limitation after acute pulmonary embolism.

Comment 4.     In line 103 : Please elaborate on what this work up required by the ESC guidelines.

Reply 4: Thank you for this comment. Patients after pulmonary embolism with at least intermediate echocardiographic probability of pulmonary hypertension according to the ESC guidelines are referred to the detailed complete work-up for CTEPH including perfusion lung scan and right heart catheterisation when indicated. Elevated BNP levels and abnormalities in the cardiopulmonary exercise test also enhance the suspicion of pulmonary hypertension, therefore in our opinion  it is worth considering to include  their assessment in the evaluation  pulmonary embolism survivors.

Comment 5.     Line 106 : Can you elaborate what echo findings you were looking at (I know these are mentioned later in the manuscript but for the reader to know preemptively what you are looking for is helpful)

Reply 5:  Thank you for this comment.  Indeed, in order to improve the manuscript, following this comment we have added following sentences (line 113 – 115) “echocardiographic criteria for suspected PH included increased systolic peak tricuspid regurgitation velocity, dilated right ventricle, flattened interventricular septum, distended inferior vena cava with diminished inspiratory collapsibility or enlarged right atrial area.”

Comment 6.     Line 77-78 “Its sensitivity in diagnosis of CTEPH is around 82% and specificity of 70%” Can you place a reference here

Reply 6: “NTproBNP sensitivity in diagnosis of CTEPH is around 82% and specificity of 70%” was written based on ERS statement on chronic thromboembolic pulmonary hypertension (Eur Respir J 2020;57:200282). Following your comment we have added reference number 7 in line 78.

Comment 7.     Line 79-81: “Electrocardiogram (ECG) abnormalities included P pulmonale, right axis deviation (RAD), right ventricle (RV) hypertrophy, right bundle branch block (RBBB) are more common in severe PH than in mild elevation of pulmonary pressure” Can you place a reference here?

Reply 7: It was written based on the 2015 ESC/ERS Guidelines for the diagnosis and treatment of pulmonary hypertension, reference number 2. Following your comment we have added this reference in line 81.

Comment 8.     Line 167 : Can you define what an elevated RV/LV ratio is ?

Reply 8: The ESC guidelines for the diagnosis and management of acute pulmonary embolism 2019 and for the diagnosis and treatment of pulmonary hypertension 2015 indicate that the parameter increasing the risk of pulmonary hypertension is right ventricle to left ventricle basal diameter ratio > 1.0. In our study, we  used the definition that elevated RV/LV ratio means more than 1.

Comment 9.     Line 168 TRPG, was not a defined abbreviation anywhere in the manuscript (except in the tables)

Reply 9: Following this recommendation we have added abbreviation of TRPG in line 122-123.

Comment 10.  Was there any difference in LV EF between the symptomatic and asymptomatic groups, I think that would be an important value to add in table 1

Reply 10: Indeed, there was a significant difference in LV EF % between the symptomatic and asymptomatic groups: 60.7 ± 5.4 % vs 62.9 ± 3.2 %; p < 0.001. We’ve added this value in table 1. , in the Results section.

Comment 11. Table 1: the word “stimulation” is misspelled.

Reply 11: Following this recommendation “stimulation” has been corrected in Table 1.

Comment 12. Table 1+2 , I am not sure that you need the extra 2 columns for “n”

Reply 12: It seemed us useful to have “n” next to each parameter as a different number of parameters were available for analysis. However, if the Reviewer would still recommend to remove “n” we are ready follow this comment.

Comment 13.  In your discussion can you comment on the theorized effect of afib and age on the difference in Pro BNP between the two groups?

Reply 13: Natriuretic peptides had been shown to be elevated in atrial fibrillation, but in our study only 10 of 155 (6.4%) symptomatic patients had atrial fibrillation, so this could not have had a significant effect on elevated BNP in the symptomatic patients. Symptomatic PE survivors were also significantly older, and BNP increases with age. We have added sentence in line 295-297: ‘Survivors with symptoms were also much older and more likely to have fibrillation than asymptomatic patients, which also explains the increased concentration of BNP’ and reference 19.

Comment 14.  In your discussion can you comment on how you came to the conclusion that their decrease in exercise capacity was sec to CTPH/PD versus other L sided issues especially since these patients had much higher rates of afib and possibly diastolic dysfunction.

Reply 14: Thank you for this comment. Symptomatic patients were first referred to echo lab. Of course,  there are many  potential causes of decreased exercise capacity including alterations of the left heart morphology and function. However, please note that the patients with at least an intermediate echocardiographic probability of pulmonary hypertension according to the ESC guidelines had a V/Q lung scan recommended. Patients with perfusion defects in lung scan were referred for a detailed diagnostic workup for CTEPH, which included right heart catheterization with selective pulmonary angiography. CTEPH was diagnosed when invasive right heart catheterization showed the mean pulmonary artery pressure (mPAP) was ≥25 mmHg at rest, pulmonary wedge pressure ≤15 mmHg, and abnormal imaging findings on the V/Q scan, while CTED was diagnosed as CTEPH however when mPAP <25 mmHg.

Comment 15. How many symptomatic patients get a R heart cath and what were the results of the ones that did?

Reply 15: All symptomatic patients with perfusion defects in lung scan (n = 20) were referred and underwent a detailed diagnostic workup for CTEPH, which included right heart catheterization with selective pulmonary angiography. CTEPH was diagnosed when invasive right heart catheterization showed the mean pulmonary artery pressure was 25 mmHg at rest, while CTED was diagnosed as CTEPH however when mPAP < 25 mmHg. There were 7 CTED cases and 13 CTEPH cases in our study.

Reviewer 2 Report

The authors have designed a study for the purpose of screening CTEPD/CTEPH in patients after PE, which was vital and necessary for the long-term management in PE patients. They found that screening for CTEPD/CTEPH should be performed in patients with reduced exercise tolerance compared to pre PE period and is not effective in asymptomatic PE survivors. However, some serious methodological and reporting issues should be addressed with extreme care.

1.The study design was unclear. Was this study a prospective or retrospective cohort study?

The population you have chosen was not defined clearly. The time range of the eligible cases were collected and the hospital you selected were necessary for a complete study design. You are supposed to make a flow chart to show the population selection.

And the definition of the outcome was also vital. You should describe the primary outcome and secondary outcome in detail.

It's significant to determine your research objectives and express your intentions clearly and apparently. Whether the purpose of your study is to predict the occurrence of CTEPH in patients after PE or make a diagnosis?

In line 93, “standard outpatient follow-up for at least 6 months following the acute PE event and were anticoagulated for at least 6 months. Briefly, we included all consecutive patients after acute PE with the exception of subjects with comorbidities.” I did not know whether there were obvious differences of the time for patients to undergo the tests of echocardiogram, ECG and NT-proBNP. You should present the time patients were included and the time patients were examined after PE accurately, which may influence the results of these indicators. Besides, the measuring times for important indicators should be presented clearly, at new-onset of PE or grouping? 

2.The presence or absence of symptoms in the patients was an important part of the article. As we know, the patients with PE have a variety of specific or unspecific symptoms. To avoid ambiguity, the precise definition and description of symptoms and the symptoms you focused was worth mentioning. There are many factors that affect activity tolerance and left heart function is one important and common factor. In your manuscript I only see the echocardiographic findings during the follow-up phase, but the patient's echocardiographic findings at the initial onset of PE and the comorbidities of patients were not found in the manuscript. Under such condition, grouping patients only based on symptoms was seemingly unreasonable and the interpretation of results was not reliable. You should show these data and make further analysis about this confounding factor, with the purpose of making your conclusion more solid.

Among other indicators related to left heart failure, except morphological indicators, the left ventricular ejection fraction was critical indicators. You have mentioned them in the section echocardiography but these results were not described in the tables.

3. Your title of the manuscript was “Electrocardiogram, echocardiogram and NT-proBNP in screening for thromboembolism pulmonary hypertension in patients after pulmonary embolism”. In this study, you have compared these indicators in patients after pulmonary embolism with and without CTEPH/CTEPD among symptomatic patients. Results of measurement of different parameters in the diagnosis of CTEPH/CTEPD in symptomatic pulmonary embolism survivors were also shown. The above was very important work.

However, I have some advice in the diagnostic tests for CTEPH/CTEPD in symptomatic pulmonary embolism survivors. You only use a single indicator to build up the ROC curves. Actually you did not make full use of your data. Why not have a try to combine several indicators for a better results?

4. You have mentioned that NT-proBNP level does not identify CTEPD from CTEPH, but I did not see the data comparison in your study. Show the data and make a solid conclusion. And you should have some discussion about this phenomenon.

The number of symptomatic patients without symptoms is only 20/155. How to explain the symptoms of the other patients? The potential reasons of this phenomenon should be discussed.

5. You have mentioned both CTED and CTEPD in the manuscript. You should express it consistently.

In line 75-78, “NT-proBNP has been evaluated to stratify risk in patients with acute PE and remains the only biomarker that seems to be a strong predictor of prognosis and therefore is widely used in the routine practice in PH centers. Its sensitivity in diagnosis of CTEPH is around 82% and specificity of 70% please. Please show the reference.

Author Response

Reviewer 2

The authors have designed a study for the purpose of screening CTEPD/CTEPH in patients after PE, which was vital and necessary for the long-term management in PE patients. They found that screening for CTEPD/CTEPH should be performed in patients with reduced exercise tolerance compared to pre PE period and is not effective in asymptomatic PE survivors. However, some serious methodological and reporting issues should be addressed with extreme care.

The study design was unclear. Was this study a prospective or retrospective cohort study?

Reply 1: This is a post hoc analysis of prospectively followed cohort of patients after pulmonary embolism. Following your comment we have added this sentence in Methods section.

The population you have chosen was not defined clearly. The time range of the eligible cases were collected and the hospital you selected were necessary for a complete study design. You are supposed to make a flow chart to show the population selection.

Reply 2: Thank you for this comment, as we mentioned in “Methods” section we included all consecutive patients after  the first episode of acute PE with the exception of subjects with comorbidities significantly limiting survival or mobility (patients with advanced cancer or bed ridden subjects), who could not be observed in the outpatient clinic. Following this comment we have clarified this issue in the Methods section.

And the definition of the outcome was also vital. You should describe the primary outcome and secondary outcome in detail.

Reply 3: The primary endpoint was the diagnosis of chronic thromboembolic disease (CTED) or chronic thromboembolic pulmonary hypertension (CTEPH)

It's significant to determine your research objectives and express your intentions clearly and apparently. Whether the purpose of your study is to predict the occurrence of CTEPH in patients after PE or make a diagnosis?

Reply 4: The aim of our study was to verify whether the echocardiographic parameters  or  electrocardiographic data  combined  with and NT-proBNP concentration  can identify  patients at risk of CTEPH and allow to improve diagnostic workup after acute PE. We tried to find a parameter indicating a suspicion of chronic occlusion of the pulmonary vessels. CTEPH was diagnosed when invasive right heart catheterization showed the mean pulmonary artery pressure was ≥ 25 mmHg at rest, while CTED was diagnosed as CTEPH however when mPAP < 25 mmHg. The results of non-invasive tests do not allow to make a diagnosis of CTEPH.

In line 93, “standard outpatient follow-up for at least 6 months following the acute PE event and were anticoagulated for at least 6 months. Briefly, we included all consecutive patients after acute PE with the exception of subjects with comorbidities.” I did not know whether there were obvious differences of the time for patients to undergo the tests of echocardiogram, ECG and NT-proBNP. You should present the time patients were included and the time patients were examined after PE accurately, which may influence the results of these indicators. Besides, the measuring times for important indicators should be presented clearly, at new-onset of PE or grouping? 

Reply 5: Patients after pulmonary embolism who continued treatment in our outpatients clinic were included to our study. Echocardiogram, ECG and NT-proBNP were assessed 6 ± 0,97 months after an acute episode of  PE. We have added this sentence in line 183.

  1. The presence or absence of symptoms in the patients was an important part of the article. As we know, the patients with PE have a variety of specific or unspecific symptoms. To avoid ambiguity, the precise definition and description of symptoms and the symptoms you focused was worth mentioning. There are many factors that affect activity tolerance and left heart function is one important and common factor.

Reply 6: We fully agree with the Reviewer. There are many factors that affect activity tolerance after PE . Indeed the  diagnosis of impaired  functional capacity  after pulmonary embolism is a  real  challenge. Symptoms and their frequency we described in detail in an earlier publication "The post-pulmonary syndrome - results of echocardiographic driven follow up after acute pulmonary embolism", which is cited in the current  manuscript.  Briefly, we assessed the presence of exertional dyspnea, effort angina, exercise limiting  palpitations,  and  a reduced exercise tolerance. According this comment we have added a following sentence in the section Methods: “All patients were evaluated for the presence of exertional dyspnea, effort angina, exercise limiting  palpitations and a reduced exercise tolerance”.

In your manuscript I only see the echocardiographic findings during the follow-up phase, but the patient's echocardiographic findings at the initial onset of PE and the comorbidities of patients were not found in the manuscript. Under such condition, grouping patients only based on symptoms was seemingly unreasonable and the interpretation of results was not reliable. You should show these data and make further analysis about this confounding factor, with the purpose of making your conclusion more solid.

Reply 7: Thank you for this comment. All the studied patients underwent echocardiographic assessment during the acute phase in order to optimize  the management of the acute period. However, please note that the comparison of the results of the echo performed in the acute phase with the  results at the time of follow up was not the scope of our study. The main goal of our study was  assess the causes of functional  impairment at follow up. So we did not attach the results of the echo study from the time of diagnosis of pulmonary embolism. Clinical evaluation, including physical performance assessment, is the first post-pulmonary diagnostic step recommended by the ESC. ECG, BPN and echo were performed in all patients included to the study, but there were few and slight abnormalities in the test of asymptomatic patients, which confirms that diagnosing patients without symptoms is ineffective.

Among other indicators related to left heart failure, except morphological indicators, the left ventricular ejection fraction was critical indicators. You have mentioned them in the section echocardiography but these results were not described in the tables.

Reply 8: We fully agree that left ventricular ejection fraction is critical indicator. Indeed, there was a significant difference in LV EF % between the symptomatic and asymptomatic groups: 60.7 ± 5.4% vs 62.9 ± 3.2 %; p < 0.001. We’ve added this value in table 1, and in the Results section. 

  1. Your title of the manuscript was “Electrocardiogram, echocardiogram and NT-proBNP in screening for thromboembolism pulmonary hypertension in patients after pulmonary embolism”. In this study, you have compared these indicators in patients after pulmonary embolism with and without CTEPH/CTEPD among symptomatic patients. Results of measurement of different parameters in the diagnosis of CTEPH/CTEPD in symptomatic pulmonary embolism survivors were also shown. The above was very important work.

However, I have some advice in the diagnostic tests for CTEPH/CTEPD in symptomatic pulmonary embolism survivors. You only use a single indicator to build up the ROC curves. Actually you did not make full use of your data. Why not have a try to combine several indicators for a better results?

Reply 9: Thank you for this suggestion. We however did not make this multivariate analysis due to  rather low  sample size of CTEPH/CTEPD group (n=20). For this sample size, we have decided that multivariate analysis should not be carried out.   

Comment 4. You have mentioned that NT-proBNP level does not identify CTEPD from CTEPH, but I did not see the data comparison in your study. Show the data and make a solid conclusion. And you should have some discussion about this phenomenon.

Reply 10: Actually, we found that BNP does not differentiate combined CTEPH + CTED from other symptomatic patients. We have added table 2a with comparison of patients after pulmonary embolism with CTEPH or with CTED among symptomatic patients.

The number of symptomatic patients without symptoms is only 20/155. How to explain the symptoms of the other patients? The potential reasons of this phenomenon should be discussed.

Reply 11: Chronic heart failure was recognized in about 50% of patients as the major cause of reduced exercise tolerance. Most of them presented preserved ejection fraction. Other causes of functional limitation in the studied group included valve heart disease (6%), coronary artery diseases (6%), chronic obstructive pulmonary diseases (6%) and newly diagnosed permanent or paroxysmal atrial fibrillation in 6.4% of patients. Noncardiopulmonary pathologies including severe obesity patients, newly diagnosed neoplasms or anemia contributed to decreased functional capacity in approximately 10% of symptomatic patients. Following your comment we have added this sentence in the section Results.

  1. You have mentioned both CTED and CTEPD in the manuscript. You should express it consistently.

Reply 12: Thank you for this comment. Following your recommendation we have added table 2a with comparison of patients after pulmonary embolism with CTEPH or with CTED among symptomatic patients.

In line 75-78, “NT-proBNP has been evaluated to stratify risk in patients with acute PE and remains the only biomarker that seems to be a strong predictor of prognosis and therefore is widely used in the routine practice in PH centers. Its sensitivity in diagnosis of CTEPH is around 82% and specificity of 70%. Please show the reference.

Reply 13: “NTproBNP sensitivity in diagnosis of CTEPH is around 82% and specificity of 70%” was written based on ERS statement on chronic thromboembolic pulmonary hypertension (Eur Respir J 2020;57:200282). Following your comment we have added reference number 7 in line 78.

Reviewer 3 Report

congratulations, interesting paper

there are no data about the cardiopulmonary comorbidities of the enrolled patients

it would be interesting also to mention  CT signs of CTED/CTEPH

Author Response

Reviewer 3:

congratulations, interesting paper, there are no data about the cardiopulmonary comorbidities of the enrolled patients; it would be interesting also to mention  CT signs of CTED/CTEPH

Reply: Thank you for your comment.

Chronic heart failure was recognized in about 50%. Most of patients presented preserved ejection fraction. Other causes of functional limitation in the studied group included valve heart disease (6%), coronary artery diseases (6%), chronic obstructive pulmonary diseases (6%) and newly diagnosed permanent or paroxysmal atrial fibrillation in 6.4% of patients. Noncardiopulmonary pathologies including severe obesity patients, newly diagnosed neoplasms or anemia contributed to decreased functional capacity in approximately 10% of symptomatic patients.

CT was not routinely performed as part of post-PE syndrome diagnostics. Symptomatic patients were first referred to echo lab. Patients with at least an intermediate echocardiographic probability of pulmonary hypertension according to the ESC guidelines had a V/Q lung scan recommended. Patients with perfusion defects in lung scan were referred for a detailed diagnostic workup for CTEPH, which included right heart catheterization with selective pulmonary angiography. CTEPH was diagnosed when invasive right heart catheterization showed the mean pulmonary artery pressure (mPAP) was 25 mmHg at rest, pulmonary wedge pressure 15 mmHg, and abnormal imaging findings on the V/Q scan, while CTED was diagnosed when mPAP <25 mmHg.

CTPA imaging has become standard of care for evaluation of patients with chronic pulmonary embolism / CTED and suspected CTEPH and it plays important role in differential diagnosis between CTED and CTEPH. 

The vascular and cardiac CT signs of CTED include :

- total vessel occlusion, partial obstruction with thrombotic materials ( wall adherent thrombus, calcified thrombus, bands, webs), poststenotic vessel dilatation

- signs of systemic collateral supply (enlargement of bronchial and nonbronchial systemic arteries)

-  features of right ventricular overload/ the indirect signs of pulmonary hypertension ( that indicate CTEPH), such as an increased main pulmonary arteries diameter, right ventricular enlargement, increased PA/aorta diameter ratio, septal bowing, and contrast reflux into the IVC and hepatic veins. 

The interstitial CTED and CTEPH signs include: scars, focal ground-glass opacities, "mosaic” pattern of attenuation with regions of low attenuation corresponding to hypoperfused segments, and bronchial anomalies. CTPA may also identify patients with CTEPH at the time of acute pulmonary embolism where CTEPH was notsuspected. It is also crucial to decide whether patients should be treated with pulmonary endarterectomy and is a sanctioned method of confirming technical success postoperatively.

https://pubmed.ncbi.nlm.nih.gov/24594422/

https://link.springer.com/article/10.1007/s00330-020-07556-4

https://err.ersjournals.com/content/26/143/160108

https://www.ncbi.nlm.nih.gov/pmc/articles/PMC4426108/
